# Evaluation of soil heavy metals pollution and the phytoremediation potential of copper-nickel mine tailings ponds

**Jianfei Shi** [1,2], **Wenting Qian**[2]*, **Zhengzhong Jin**[2]*, **Zhibin Zhou**[2], **Xin Wang**[1], **Xiaoliang Yang**[1]

1 University of Chinese Academy of Science, Beijing, China, 2 Xinjiang Institute of Ecology and Geography, Chinese Academy of Science/National Native-Oasis Ecology Construction Engineering Technology Research Center, Urumqi, China

* jinzz@ms.xjb.ac.cn (ZJ); qwt@ms.xjb.ac.cn (WQ)

**Data Availability Statement:** All relevant data are within the paper.

**Funding:** This work was funded by the National Key Research and Development Program of China

## Abstract

Heavy metal pollution in soils caused by mining has led to major environmental problems around the globe and seriously threatens the ecological environment. The assessment of heavy metal pollution and the local phytoremediation potential of contaminated sites is an important prerequisite for phytoremediation. Therefore, the purpose of this study was to understand the characteristics of heavy metal pollution around a copper-nickel mine tailings pond and screen local plant species that could be potentially suitable for phytoremediation. The results showed that Cd, Cu, Ni, and Cr in the soil around the tailings pond were at the heavy pollution level, Mn and Pb pollution was moderate, and Zn and As pollution was light; The positive matrix factorization (PMF) model results showed that the contributions made by industrial pollution to Cu and Ni were 62.5% and 66.5%, respectively, atmospheric sedimentation and agricultural pollution contributions to Cr and Cd were 44.6% and 42.8%, respectively, the traffic pollution contribution to Pb was 41.2%, and the contributions made by natural pollution sources to Mn, Zn, and As were 54.5%, 47.9%, and 40.0% respectively. The maximum accumulation values for Cu, Ni, Cr, Cd, and As in 10 plants were 53.77, 102.67, 91.10, 1.16 and 7.23 mg/kg, respectively, which exceeded the normal content of heavy metals in plants. *Ammophila breviligulata* Fernald had the highest comprehensive extraction coefficient (*CEI*) and comprehensive stability coefficient (*CSI*) at 0.81 and 0.83, respectively. These results indicate that the heavy metal pollution in the soil around the copper nickel mine tailings pond investigated in this study is serious and may affect the normal growth of plants. *Ammophila breviligulata* Fernald has a strong comprehensive remediation capacity and can be used as a remediation plant species for multiple metal compound pollution sites.

## 1. Introduction

Mining causes damage to regional vegetation [1, 2] and the tailings from ore refining are an important source of environmental pollution [3, 4]. Tailings are residual wastes from the

(2018YFC1802903). The funders had no role in study design, data collection and analysis, decision to publish, or preparation of the manuscript.

**Competing interests:** The authors have declared that no competing interests exist.

processing and production of ores and industrial minerals and contain unstable primary and secondary minerals [5]. It is estimated that more than 10 billion tons of tailings are produced in the world every year and a large amount of tailings accumulate leading to a large environmental footprint in time and space [6]. Generally, tailings are acidic, mainly composed of silt or sand sized particles [7, 8]. In arid climates, the fine particles on the tailings surface are rich in pollutants and are vulnerable to influence the wind-induced dispersion and hydraulic action [7, 9]. This means that they continue to diffuse to the surrounding environment, eventually leading to expansion of the polluted area. Therefore, soil heavy metal pollution in arid areas may pose a greater threat to the ecological environment than in other areas, such as humid or semi-humid areas.

Heavy metals are not affected by biodegradation, but are converted between various species, which means that it is very difficult to remediate heavy metals in soil [10]. In recent decades, researchers have tried to develop effective remediation technologies for contaminated soils [11]. Phytoremediation methods have attracted extensive attention due to their advantages, such as being solar energy driven, their cost-effectiveness, and they are environmentally friendly [10]. Hyperaccumulative plants, such as *Sedum alfredii*, *Pteris vittata*, and *Lolium perenne*, and heavy metals tolerant plants with large biomasses, such as *Ricinus communis*, *Cannabis sativa*, and *Pistacia lentiscus*, which can accumulate large amounts of heavy metals, have been the focus of many studies [12]. However, in the arid areas of northwest China, climate, water, and soil constraints often mean that it is difficult to use these non-native plants to remediate heavy metal pollution in soils. Therefore, it is necessary to evaluate the heavy metal remediation potential of native plants in order to screen plant species suitable for the remediation of heavy metal pollution in soil in arid areas.

The biological accumulation factor (*BAF*) and the biological concentration factor (*BCF*) are important parameters used to evaluate the potential phytoremediation of heavy metals in soil. Wang et al. [13] screened some plant species for As, Pb, and Sb remediation based on their *BCF* and Marrugo-Negrete et al. [14] screened plant species for Hg pollution based on their *BAF* and *BCF*. Many researchers have used the *BAF* and *BCF* to screen plant species for their potential remediation of heavy metal pollution in soil. However, the *BAF* or *BCF* only show the remediation potential of plants for a single heavy metal and do not reflect the comprehensive remediation potential of plants for multiple metals. Therefore, based on the *BAF* and *BCF*, this study proposes using the comprehensive extraction coefficient (*CEI*) and the comprehensive stability coefficient (*CSI*) to quantitatively evaluate the main mechanisms (phytoextraction and phytostabilization) associated with phytoremediation during soil heavy metal remediation, and provides a new plant screening method for the phytoremediation of multi-metal compound pollution sites.

In summary, the aims of this study were (1) to explore the impact of tailings accumulation on the enrichment of heavy metals in the surrounding soil, (2) to identify the potential sources of heavy metals using the PMF model, (3) to establish a new evaluation method for phytoremediation potential, and (4) to screen the native remediation species suitable for multiple heavy metal compound pollution sites in arid areas.

## 2. Materials and methods

### 2.1 Study area description and sample collection

The copper-nickel mine tailings pond was built in 1999 and is located in northern Xinjiang Province and in the eastern part of Fuyun County (Fig 1). The design level of the pond capacity is grade V and the floor area is about $1.5 \times 10^5$ m$^2$. The slag waste is disposed of by the traditional wet discharge method and the discharge volume is about 1,000 t/d. The altitude of the

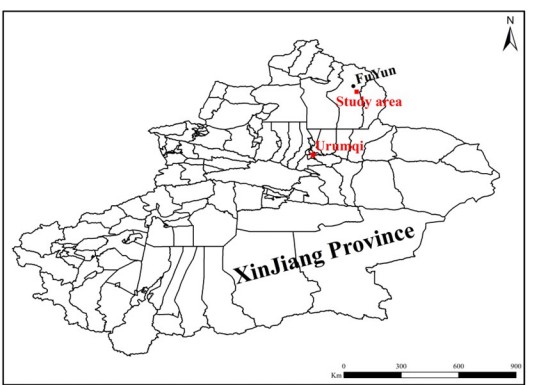 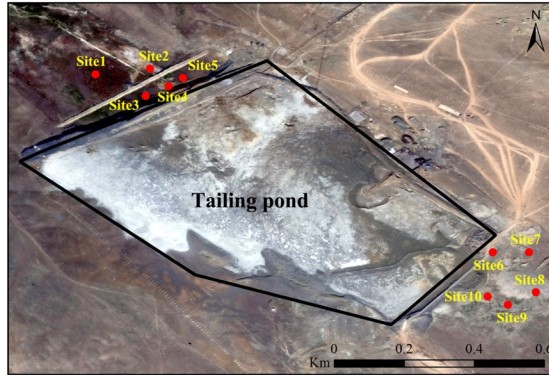

**Fig 1. Location of the study area and distribution of the sample points.** The map is based on the standard map number GS (2016) 1666 downloaded from the standard map service website of the National Bureau of Mapping Geographic Information, and the base map is not modified. https://earthexplorer.usgs.gov/.

area ranges from 500–900 m and the soil types are mainly light brown calcium soil and typical brown calcium soil. The area is subjected to a continental cold temperate climate. It is dry and windy in spring, but cold in winter. There are also large temperature differences between day and night. The annual average temperature is 3.0°C, the annual precipitation is 186.4 mm, the annual evaporation is 1,829.7 mm, the annual extreme maximum temperature is 42.2°C, and the extreme minimum temperature is –51.5°C. It is one of the high, cold regions in China. In September, 2021, the natural plants around the tailings pond were investigated and sampled. During this survey, 10 native plants belonging to eight families were collected and recorded (Table 1). The plant samples were then divided into underground and aboveground parts and each plant was replicated three times. Soil samples from around the plant roots were collected at the same time as the plant samples. The soil sampling depth was 0–20 cm and a total of 60 plant samples and 30 soil samples were collected.

## 2.2 Sample processing and determination

The plant samples were first washed with tap water and then three times with deionized water. They were placed in an oven at 105°C for 20 minutes, dried at 70°C to a constant weight, ground with a grinder to pass through a 100 mesh nylon sieve, and bagged. Then, 0.2 g sub-samples of the plants were weighed out, placed in a microwave digester, and digested in an $HNO_3$-$H_2O_2$ digestion system ($HNO_3$:$H_2O_2$ = 5:1, volume ratio) until the liquid clarified. The

**Table 1. Composition of native plant species around the tailing compounds.**

| Species | Site | Family | Life form |
|---|---|---|---|
| *Polygonum aviculare* L. | 9 | Polygonaceae | Annual grass |
| Salsola ruthenica | 6 | Chenopodiaceae | |
| *Halogeton arachnoideus* Moq. | 10 | Amaranthaceae | |
| *Atriplex patens* | 8 | Chenopodiaceae | |
| *Klasea centauroide*s subsp. *polycephala* (Iljin) L. Martins | 4 | Compositae | Perennial grass |
| *Limonium sinense* (Girard) Kuntze | 1 | Plumbaginaceae | |
| *Sphaerophysa salsula* (Pall.) DC. | 2 | Leguminosae | |
| *Ammophila breviligulata* Fernald | 5 | Gramineae | |
| *Phragmites australis* (Cav.) Trin. ex Steud | 3 | Gramineae | |
| *Peganum harmala* L. | 7 | Zygophyllaceae | |

soil samples were dried after removing impurities, such as stones and animal and plant residues. These samples were then ground with agate mortar through a 200 mesh nylon sieve, and bagged until needed. Then, 0.1 g subsamples of the soil were digested in a microwave digester under a $HNO_3$-HF-HCl digestion system ($HNO_3$: HF: HCl = 3:1:1, volume ratio) until the liquid clarified. The experiment was verified using the blank control method, the double parallel sample method, and the standard addition recovery method to ensure the accuracy of the experimental and determination processes. The standard samples used for the determination of plant samples and soil samples are GBW 10052a (GSB-30a) and GBW07426 (GSS-12). The standard curve is prepared with mixed standard solution and diluted with 2% dilute nitric acid. The supernatants from the subsamples were removed and the contents of eight heavy metals (Cr, Ni, Cu, Zn, Cd, Pb, Mn, As) were determined by inductively coupled plasma mass spectrometry (ICP-MS).

## 2.3 Evaluation of heavy metal pollution in soil

**2.3.1 Single factor pollution index method.** The single factor pollution index ($P_i$) is a common method used to evaluate the degree to which a soil has been polluted with heavy metals [15]. The index is used to evaluate environmental quality by comparing the measured value with the standard value. The calculation formula is as follows [15, 16]:

$$P_i = C_i/S_i \tag{1}$$

where $P_i$ is the single-component contamination index, $C_i$ is the measured concentration of examined metal $i$ in the soil, and $S_i$ is the background concentration of metal $i$. This study took the background value for heavy metals in Xinjiang soil as the standard value. The evaluation results were divided into five grades: $P_i \leq 0.7$, safe; $0.7 < P_i \leq 1.0$, warning; $1 < P_i \leq 2$, slight pollution; $2 < P_i \leq 3$, moderate pollution; and $P_i > 3$, heavy pollution.

**2.3.2 Nemerow comprehensive pollution index method.** The Nemerow comprehensive pollution index is used to calculate the comprehensive pollution effects of all assessed heavy metals. It can comprehensively reflect the effects of various heavy metals on soil and avoids the weakening of heavy metal weights caused by averaging. The calculation formula is as follows [17–19]:

$$P_{\text{com}} = \sqrt{\frac{P_{\text{max}}^2 + P_{\text{ave}}^2}{2}} \tag{2}$$

where $P_{\text{com}}$ is the composite contamination index, $P_{\text{ave}}$ is the average value of the single-factor index, and $P_{\text{max}}$ is the maximum value of the single-factor index. The evaluation results were divided into five grades: $P_{\text{com}} \leq 0.7$, safe; $0.7 < P_{\text{com}} \leq 1.0$, warning; $1 < P_{\text{com}} \leq 2$, light pollution; $2 < P_{\text{com}} \leq 3$, moderate pollution; and $P_{\text{com}} > 3$, heavy pollution.

## 2.4 Evaluation of phytoremediation potential

**2.4.1 Enrichment characteristics of heavy metals in plants.** Heavy metals absorbed by plants are mainly characterized by the biological accumulation factor ($BAF$) and the biological concentration factor ($BCF$). Their calculation formulas are as follows [20]:

$$BAF_i = C_{shoot}^i/C_{soil}^i \tag{3}$$

$$BCF_i = C_{root}^i/C_{soil}^i \tag{4}$$

where $i$ represents the ith heavy metal element; $BAF_i$ is the enrichment coefficient of plants for

the *ith* heavy metal, which is a parameter that is used to evaluate the ability of plant stems and leaves to extract heavy metals from soil; $BCF_i$ is the stability coefficient of plants for the *ith* heavy metal and is a parameter used to evaluate the ability of plant roots to stabilize heavy metals; $C_{shoot}^i$ is the content of the *ith* heavy metal in the aboveground part of the plant; $C_{root}^i$ is the content of the *ith* heavy metal in the underground part of the plant; and $C_{soil}^i$ is the content of the *ith* heavy metal in the plant growth matrix.

**2.4.2 Comprehensive extraction index for plants.** The comprehensive extraction index (*CEI*) for plants is based on fuzzy synthesis and can be used to evaluate the plant comprehensive extraction potential for various heavy metals under multiple heavy metal combined pollution conditions. The calculation formula is as follows [21, 22]:

$$CEI = \left(\frac{1}{n}\right)\sum_1^n u_i \tag{5}$$

$$u_i = \begin{cases} 0 & BAF = BAF_{min} \\ \dfrac{BAF - BAF_{min}}{BAF_{max} - BAF_{min}} &, \quad BAF_{min} < BAF < BAF_{max} \\ 1 & BAF = BAF_{max} \end{cases} \tag{6}$$

where *CEI* is the comprehensive extraction index for plants; *n* is the total number of heavy metals analyzed; *BAF* is the biological accumulation factor of plants for the *ith* heavy metal; $BAF_{max}$ and $BAF_{min}$ are the maximum and minimum values of the biological accumulation factor for *ith* heavy metals in the investigated plants, respectively; and $u_i$ is the fuzzy membership value. The comprehensive extraction potential of plants is divided into three grades, poor (*CEI*≤0.4), good (0.4<*CEI*<0.7), and excellent (*CEI*≥0.7).

**2.4.3 Plant comprehensive stability index.** The plant comprehensive stability index (*CSI*) is based on fuzzy synthesis and can be used to evaluate the plant root comprehensive stability potential for various heavy metals under multiple heavy metal combined pollution conditions. The calculation formula is as follows:

$$CSI = \left(\frac{1}{n}\right)\sum_1^n u_i \tag{7}$$

$$u_i = \begin{cases} 0 & BCF = BCF_{min} \\ \dfrac{BCF - BCF_{min}}{BCF_{max} - BCF_{min}} &, \quad BCF_{min} < BCF < BCF_{max} \\ 1 & BCF = BCF_{max} \end{cases} \tag{8}$$

where *CSI* is the comprehensive stability index for a plant; *n* is the total number of heavy metals analyzed; *BCF* is the biological accumulation factor of plants for the *ith* heavy metal; $BCF_{max}$ and $BCF_{min}$ are the maximum and minimum values of the biological concentration factor for *ith* heavy metals in the investigated, plants, respectively; and $u_i$ is the fuzzy membership value. The comprehensive stability potential of plants is divided into three grades, poor (*CSI*≤0.4), good (0.4<*CSI*<0.7), and excellent (*CSI*≥0.7).

## 2.5 Data processing and analysis

IBM SPSS statistics 22.0 (IBM Corp., Armonk, NY, USA) was used to statistically analyze the heavy metal contents in the soil, EPA PMF 5.0 was used for the source analysis of heavy metals

**Table 2. Statistics for heavy metal content in the soil around the tailing reservoir area (n = 30).**

| Heavy metal | Cr | Ni | Cu | Zn | Cd | Pb | Mn | As |
|---|---|---|---|---|---|---|---|---|
| Mean(mg/kg) | 529.0 | 1168.0 | 738.5 | 118.2 | 6.4 | 40.1 | 1600.3 | 13.9 |
| Minimum(mg/kg) | 255.2 | 393.6 | 307.9 | 99.4 | 2.7 | 18.1 | 1318.5 | 5.4 |
| Maximum(mg/kg) | 911.3 | 2861.1 | 1654.3 | 151.6 | 12.7 | 69.6 | 1810.8 | 24.9 |
| Coefficient of Variation (%) | 47.19 | 61.72 | 53.16 | 11.87 | 37.94 | 43.25 | 9.38 | 38.22 |
| BG$_1$[a] (mg/kg) | 49.3 | 26.6 | 26.7 | 68.8 | 0.12 | 19.4 | 688 | 11.2 |
| BG$_2$[b] (mg/kg) | 200 | 100 | 100 | 250 | 0.3 | 120 | - | 30 |

[a]BG$_1$: background values for heavy metals in Xinjiang soil.

[b]BG$_2$: Soil environment quality risk control standard for the contamination of agricultural land.

(GB 15618–2018).

in soil, and Origin 2020 (OriginLab Corporation, Northampton, MA, USA) and ArcGIS 10.6 (ESRI Inc., Redlands, CA, USA) were used for the graphics.

# 3. Results and analysis

## 3.1 Statistical characteristics of soil heavy metal content

According to the descriptive statistical analysis (Table 2), the Cr, Ni, Cu, Zn, Cd, Pb, Mn, and As contents in the soil samples were 255.2–911.3, 393.6–2861.1, 307.9–1654.3, 99.4–151.6, 2.7–12.7, 18.1–69.6, 1318.5–1810.8, and 5.4–4.9 mg/kg, with average contents of 529.0, 1168.0, 738.5, 118.2, 6.4, 40.1, 1600.3, and 13.9 mg/kg, respectively. The average Cr, Ni, Cu, Zn, Cd, Pb, Mn, and As contents in the soil samples were 10.7, 43.9, 27.7, 1.7, 53.3, 2.1, 2.3, and 1.2 times more than the background values for heavy metals in Xinjiang soils, respectively. Compared to the soil environment quality risk control standard (GB 15618–2018), Cr, Ni, Cu, and Cd in the soil samples exceeded the standard by about 2.6, 11.7, 7.4, and 21.3 times, respectively. The variation indexes for Ni and Cu in the soil were 61.72% and 53.16% respectively, indicating that Ni and Cu are causing serious localized pollution. The variation indexes for Cr, Cd, Pb, and As were 47.19%, 37.94%, 43.25%, and 38.22%, respectively, which showed that the variation indexes were relatively weak. The variation indexes for Zn and Mn were 11.87% and 9.38%, respectively, which are less than 25% and shows that the variation was low and that they are less affected by external conditions.

## 3.2 Evaluation of heavy metal pollution in soil

The pollution caused by different heavy metal elements in the soil was evaluated using Formula (1). The results (Fig 2) show that the single factor pollution indexes for heavy metal elements Cd, Cu, Ni, Cr, Mn, Pb, Zn, and As were between 22.5–108.5, 11.5–62.0, 14.8–107.6, 1.7–6.1, 1.9–2.6, 0.9–3.6, 1.4–2.2, and 0.5–2.2, respectively. Among them, the average single factor pollution indexes for Cd, Cu, Ni, and Cr were 53.2, 27.7, 43.9, and 3.5, respectively, which were greater than 3 and meant that they reached the heavy pollution grade; the single factor pollution index mean values for Mn and Pb were 2.3 and 2.1, respectively, which were at the moderate pollution level; and the single factor pollution index mean values for Zn and As were 1.7 and 1.2, respectively, which were at the light pollution level. The Nemerow comprehensive pollution index results, calculated using Formula (2) (Fig 2), showed that the Nemerow pollution indexes for all sample points ranged from 25.0 to 79.0, with an average of 43.9, which was at the heavy pollution level. In general, the soil was heavily polluted with metals and remediation measures were urgently needed.

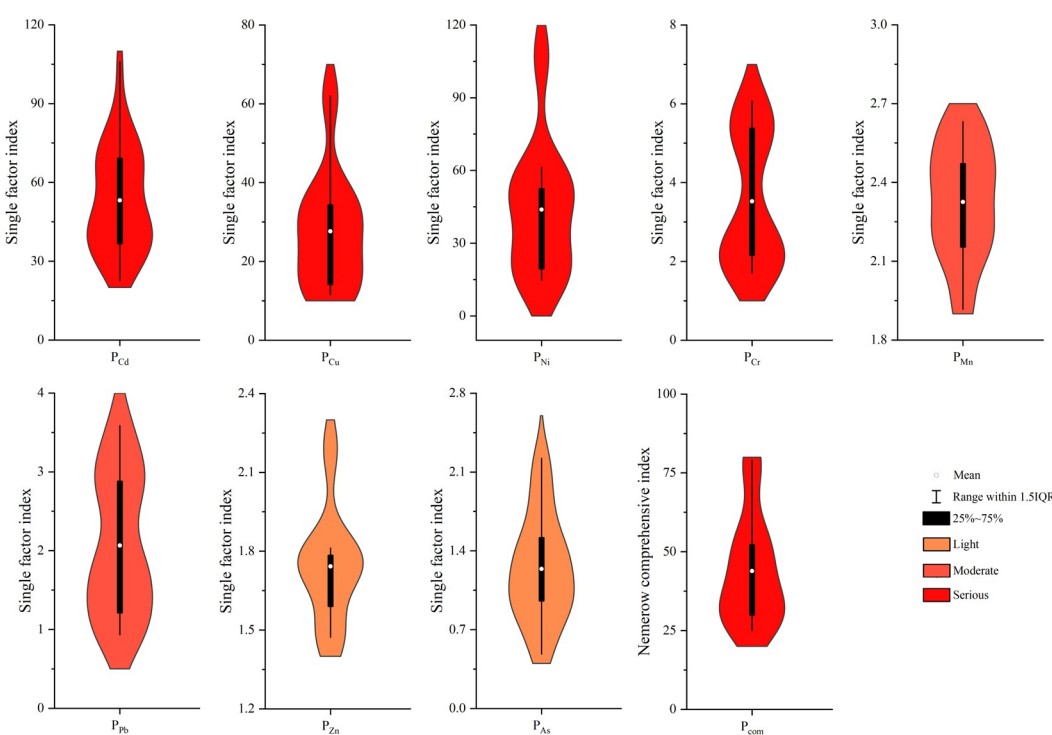

**Fig 2. Descriptive statistics for the heavy metal pollution indexes.**

### 3.3 Source analysis of heavy metals in soil

The content data for eight heavy metals in 30 samples, and the uncertainty data related to these contents, were used as the input files for the PMF 5.0 model. A total of four factors were determined as the optimal solution to ensure the rationality of the model. Most of the residuals ranged from –3 to 3 and the signal-to-noise ratios (S/N) of the eight selected heavy metals were greater than 2. Factor 1 accounted for 21.13% of the total contribution. Cd and Cr had the highest loads at 44.6% and 42.8%, respectively. The correlation analysis results showed that the correlation coefficient between Cd and Cr was 0.71, reaching the very significant level ($P < 0.01$). This indicated that factor 1 was the main pollution source for Cd and Cr in the soil. Factor 2 accounted for 30.66% of the total contribution and Cu and Ni had the highest loads at 62.5% and 66.5%, respectively. The correlation analysis results showed that the correlation coefficient between Cu and Ni was 0.88, reaching the very significant level ($P < 0.01$), which indicated that factor 2 was the main pollution source for Cu and Ni in the soil. Factor 3 accounted for 26.96% of the total contribution and Mn, As, and Zn had high contribution rates of 54.5%, 47.9%, and 40.0% respectively. Factor 4 accounted for 21.25% of the total contribution and Pb had the highest load at 41.2%, which indicated that factor 4 was the main source of Pb pollution in the soil.

### 3.4 Heavy metal content in plants

The heavy gold contents in different plants around the tailings pond are shown in Fig 3. It showed that the heavy metal levels in different plants or different parts of the same plant showed large variations, which may be closely related to the physical and chemical properties of the soil, the occurrence forms of heavy metals, and the characteristics of the plants themselves. The contents of eight heavy metals (Cr, Ni, Cu, Zn, Cd, Pb, Mn, and As) in the

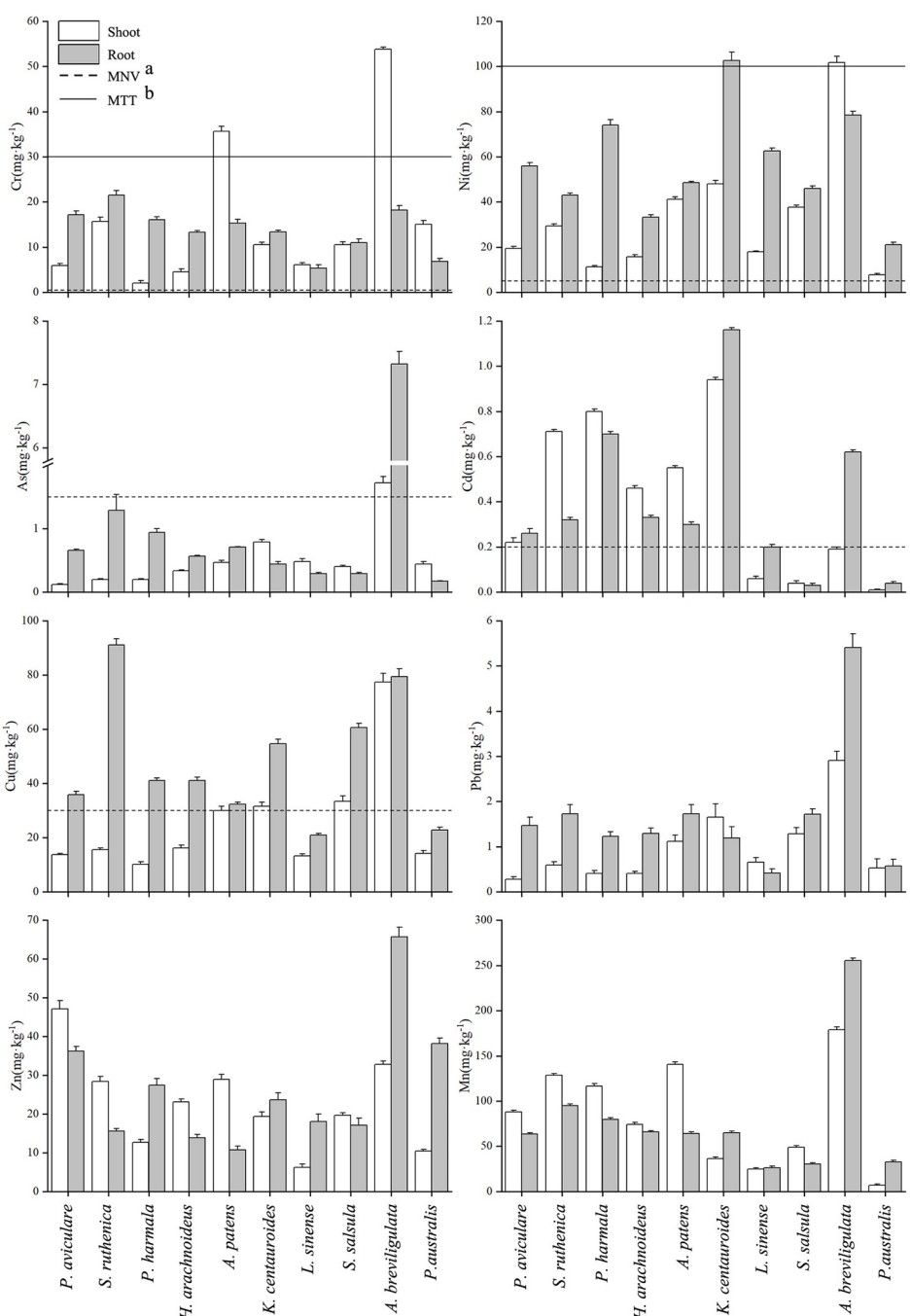

**Fig 3. Heavy metal contents in the aboveground and underground parts of 10 native plants (a: MNV, Maximum normal value, b: MTT, Maximum toxicity threshold).**

aboveground parts of 10 plants varied from 2.02–53.77, 7.75–101.67, 10.22–77.37, 6.19–47.13, 0.01–0.94, 0.28–2.91, 24.57–178.67, and 0.12–1.72 mg/kg, respectively, and the Cr, Ni, Cu, Zn, Cd, Pb, Mn, and As variation ranges for their contents in the underground parts of the 10 plants were 5.34–21.50, 21.00–102.67, 20.97–91.10, 10.73–65.73, 0.03–1.16, 0.42–5.41, 26.33–255.33, and 0.17–7.3 mg/kg, respectively. In general, the Ni, Cu, Pb, and As contents in the underground parts of most plants were higher than those in the aboveground part, while Cr,

Mn, and Cd were lower than those in the aboveground parts. The heavy metal contents in plants from high to low were Mn > Ni > Cu > Zn > Cr > Pb > As > Cd. Compared to the normal range of heavy metals in plants [23], the heavy metal elements Ni (Ni: 0.1–5 mg/kg) and Cr (Cr: 0.1–0.5 mg/kg) in the underground or aboveground parts of all plants exceeded the maximum value for the normal content range. The Cu levels in the underground parts of other plants exceeded the maximum normal level (Cu: 5–30 mg/kg), except *P. australis* and *L. sinense*. The Cd levels in the plants (aboveground or underground parts) exceeded the maximum for the normal range (Cd: 0.01–0.2 mg/kg), except for *L. sinense*, *S. salsula* and *P. australis*. The As content in all plants (aboveground and underground parts), except for *A. breviligulata*, did not exceed the maximum normal range (As: 1–1.5 mg/kg), and the Mn, Zn, and Pb contents in all plants (aboveground and underground parts) were within the normal range (Mn: 30–300 mg/kg, Zn: 25–250 mg/kg, and Pb: 5–10 mg/kg). Compared with the toxicity threshold range in plants [23], the Cr content in the aboveground parts of *A. patens* and *A. breviligulata* exceeded the maximum toxicity threshold (Cr: 5–0.5 mg/kg). The Ni content in the underground part of *K. centauroides* and the aboveground part of *A. breviligulata* also exceeded the maximum toxicity threshold range (Ni: 10–100 mg/kg). The contents of As, Cd, Pb, Zn and Mn in all plants (aboveground or underground) were lower than the maximum value of the toxicity threshold range (As: 5–20 mg/kg; Cd: 5–30 mg/kg; Pb: 30–300 mg/kg; Zn: 100–400 mg/kg, Mn: 400–100 mg/kg).

## 3.5 Heavy metal absorption characteristics of plants

The biological accumulation factor (*BAF*) shows the ability of plant stems and leaves to accumulate heavy metals, which means that it reflects the ability of plants to remove heavy metals from soil. Table 3 shows that there were obvious differences in the accumulation capacity of plants for heavy metals. It also shows the differences among different plants for the same element and in the same plant for different elements. The minimum biological accumulation factors for the 10 plants around the tailings pond for Cr, Mn, Ni, Cu, Zn, As, Cd, and Pb are 0.006, 0.003, 0.014, 0.051, 0.010, 0.001, and 0.008, respectively, and the maximum values are 0.110, 0.113, 0.104, 0.107, 0.457, 0.133, 0.190, and 0.074, respectively. Among the 10 plants, *A. patens* has a strong ability to extract Cr, *Polygonum aviculare L.* as a strong ability to extract Zn, *S. ruthenica* has a strong ability to extract Cd, and *A. breviligulata* shows a strong extraction ability for Mn, Ni, Cu, As, and Pb. The biological concentration factor (*BCF*) can reflect the ability of plants to stabilize heavy metals in soil. Table 4 shows that the minimum plant stability coefficients for the 10 plants around the tailings pond for Cr, Mn, Ni, Cu, Zn, As, Cd,

**Table 3. Biological accumulation factors *(BAF)* for native plants around the tailings pond.**

| Species | BAF | | | | | | | |
|---|---|---|---|---|---|---|---|---|
| | Cr | Mn | Ni | Cu | Zn | As | Cd | Pb |
| *P. aviculare* | 0.012 | 0.049 | 0.014 | 0.023 | 0.457 | 0.01 | 0.05 | 0.008 |
| *S. ruthenica* | 0.048 | 0.087 | 0.074 | 0.017 | 0.238 | 0.019 | 0.19 | 0.032 |
| *P. harmala* | 0.006 | 0.08 | 0.022 | 0.032 | 0.084 | 0.035 | 0.176 | 0.02 |
| *H. arachnoideus* | 0.015 | 0.044 | 0.022 | 0.042 | 0.198 | 0.039 | 0.083 | 0.017 |
| *A. patens* | 0.11 | 0.078 | 0.04 | 0.054 | 0.284 | 0.042 | 0.109 | 0.031 |
| *K. centauroides* | 0.014 | 0.023 | 0.03 | 0.039 | 0.158 | 0.049 | 0.097 | 0.029 |
| *L. sinense* | 0.008 | 0.019 | 0.013 | 0.016 | 0.051 | 0.031 | 0.007 | 0.011 |
| *S. salsula* | 0.012 | 0.031 | 0.027 | 0.02 | 0.161 | 0.027 | 0.005 | 0.019 |
| *A. breviligulata* | 0.106 | 0.113 | 0.104 | 0.107 | 0.276 | 0.133 | 0.03 | 0.074 |
| *P. australis* | 0.018 | 0.017 | 0.003 | 0.014 | 0.084 | 0.022 | 0.001 | 0.01 |

**Table 4. Biological concentration factors (*BCF*) for native plants around the tailings pond.**

| Species | BCF | | | | | | | |
|---|---|---|---|---|---|---|---|---|
| | Cr | Mn | Ni | Cu | Zn | As | Cd | Pb |
| *P. aviculare* | 0.036 | 0.035 | 0.04 | 0.06 | 0.351 | 0.056 | 0.059 | 0.043 |
| *S. ruthenica* | 0.066 | 0.064 | 0.109 | 0.099 | 0.131 | 0.121 | 0.086 | 0.094 |
| *P. harmala* | 0.048 | 0.055 | 0.143 | 0.127 | 0.182 | 0.164 | 0.154 | 0.061 |
| *H. arachnoideus* | 0.045 | 0.039 | 0.047 | 0.108 | 0.118 | 0.065 | 0.06 | 0.055 |
| *A. patens* | 0.047 | 0.036 | 0.047 | 0.059 | 0.105 | 0.064 | 0.06 | 0.047 |
| *K. centauroides* | 0.019 | 0.042 | 0.063 | 0.067 | 0.193 | 0.027 | 0.119 | 0.021 |
| *L. sinense* | 0.007 | 0.02 | 0.046 | 0.026 | 0.15 | 0.018 | 0.022 | 0.007 |
| *S. salsula* | 0.012 | 0.019 | 0.033 | 0.037 | 0.141 | 0.02 | 0.004 | 0.025 |
| *A. breviligulata* | 0.036 | 0.162 | 0.081 | 0.109 | 0.553 | 0.567 | 0.097 | 0.138 |
| *P. australis* | 0.008 | 0.019 | 0.007 | 0.022 | 0.309 | 0.008 | 0.005 | 0.011 |

and Pb were 0.007, 0.019, 0.007, 0.022, 0.105, 0.008, 0.004, and 0.007, respectively, and the maximum values were 0.066, 0.162, 0.143, 0.127, 0.553, 0.567, 0.154, and 0.138, respectively. Among them, *S. ruthenica* can strongly stabilize Cr, *P. harmala* can strong stabilize Ni, Cu, and Cd, and *A. breviligulata* can strongly stabilize Mn, Zn, As, and Pb.

### 3.6 Comprehensive evaluation of phytoremediation potential

The plant comprehensive extraction index and plant comprehensive stability index results, calculated based on Fuzzy evaluation (Fig 4), showed that among the 10 plants investigated, the plant comprehensive extraction indexes for *P. australis*, *L. sinense*, *S. salsula*, *H. arachnoideus*, *K. centauroides*, *P. harmala*, and *S. ruthenica* were 0.04, 0.05, 0.14, 0.23, 0.25, 0.26, and 0.30, respectively, which were less than 0.4, which meant that their comprehensive removal potentials for heavy metals were poor. The plant comprehensive extraction index for *A. patens* was 0.52, which meant that its comprehensive removal potential for heavy metals was good; and the plant comprehensive extraction index for *A. breviligulata* was 0.83, which means that its comprehensive removal potential level for heavy metals was excellent, indicating that it had a strong comprehensive removal ability for heavy metals. The plant comprehensive stability indexes for *P. australis*, *L. sinense*, *S. salsula*, *A. patens*, *K. centauroides*, *P. aviculare*, and *H. arachnoideus* were smaller at 0.07, 0.07, 0.08, 0.28, 0.29, 0.31, and 0.34, respectively, which are all less than 0.4. Therefore, their potential stability indexes for heavy metals are poor, indicating that their comprehensive ability to stabilize heavy metals is weak. The plant comprehensive stability indexes for *S. ruthenica* and *P. harmala* were 0.53 and 0.60 respectively, which meant that their potential stability grades for heavy metals were good. The plant comprehensive stability index for *A. breviligulata* was 0.81, which meant that its stability potential index for heavy metals was excellent, indicating that it had strong comprehensive ability to stabilize heavy metals. Therefore, *A. breviligulata* had greater remediation potential for heavy metals in soil than the other plants.

## 4. Discussion

### 4.1 Soil heavy metal pollution characteristics and source analysis

Factor 1 is the main source of Cr and Cd, contributing 44.6% and 42.8%, respectively (Fig 5). The Nemerow comprehensive pollution index evaluation results show that the soil is heavily polluted with Cr and Cd (Fig 2). In this study, Cd and Cr reached a very significant level ($P < 0.01$), indicating that there may be similar sources of Cr and Cd pollution in the study

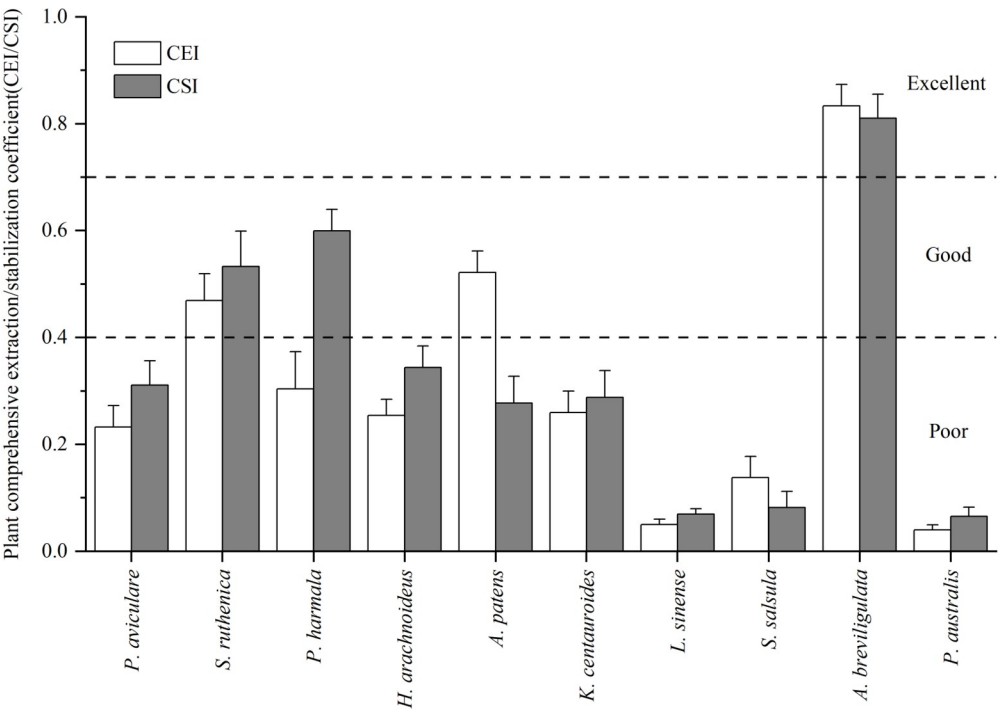

**Fig 4. Plant comprehensive enrichment/stability coefficients for the 10 native plants.**

area. Atmospheric deposition is an important source of Cr and Cd in soil. The annual Cd and Cr inputs into soil via atmospheric deposition is 493 and 7392 t/a, respectively [24]. Wang et al. [25] showed that the soil in Jiangsu Province was seriously polluted with Cr and Cd and that agricultural and industrial activities were the main reasons for Cr and Cd the enrichment in the soils. Relevant research also showed that chemical fertilizer, livestock manure, and irrigation water are important sources of Cd and Cr [24, 26, 27]. Guo et al. [28] showed that agricultural activities made large contributions to Cd and Cr pollution at 41.63% and 60.05%, respectively. There are many mining plants in Fuyun County, China, including copper nickel

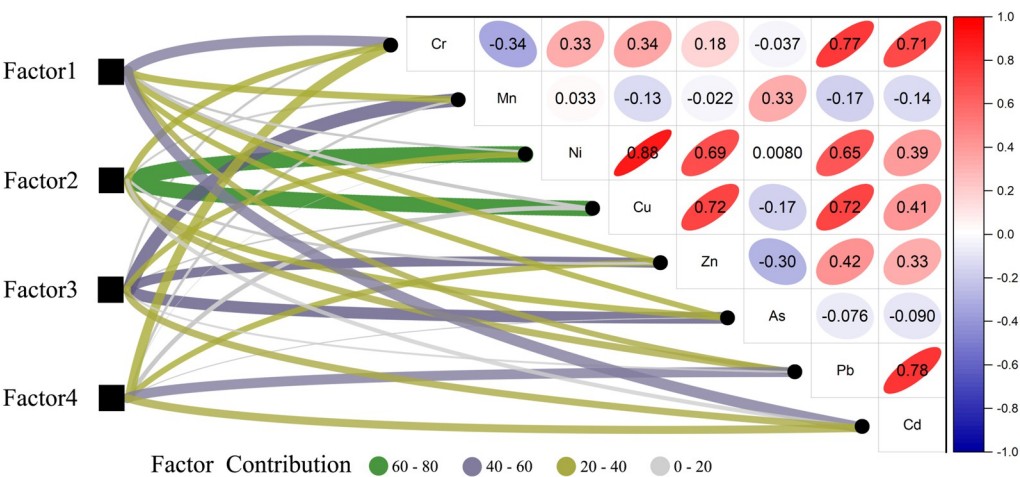

**Fig 5. Correlation analysis and source analysis of the heavy metals.**

ore, iron ore, gold mine, and rare metal mines. The pollution of heavy metals Cr and Cd produced in these industrial activities may enter the soil through atmospheric deposition. In addition, the sampling point is located in the important pasture area, and the animal husbandry activities are frequent. A large amount of feces from cattle, sheep and horses may also be the cause of heavy metal Cr and Cd pollution in the study area. Therefore, factor 1 is considered to be a composite pollution source that includes atmospheric sedimentation and agricultural pollution sources.

Factor 2 made the greatest contribution to Cu and Ni levels at 62.5% and 66.5%, respectively. The Nemerow comprehensive pollution index evaluation results show that the soil is heavily polluted with both Cu and Ni (Fig 2). Previous studies have shown that Cu and Ni in copper nickel ore tailings are serious pollutants [29]. The correlation analysis shows that the Cu and Ni in the soil were highly correlated ($P < 0.01$), which indicated that they may have the same pollution source. Li et al. [30]. found that there were many types of heavy metal pollution in the soil around the copper nickel mine tailings pond, among which Cu and Ni were the main pollutants in the soil. Their results were similar to the results produced by this study. Kabala et al. [31] showed that the Cu concentration in the soil taken from the east of the copper mine tailings pond was negatively related to the distance from the tailings pond and suggested that the tailings pond may be the main source of Cu pollution in the soil. As the sampling point was located near the copper nickel ore smelting plant and copper nickel ore tailings pond, factor 2 is considered to be an industrial pollution source.

Factor 3 made a considerable contribution to Mn, As, and Zn levels at 54.5%, 47.9%, and 40.0%, respectively. The Nemerow comprehensive pollution index evaluation results showed that Mn was moderately polluting, whereas As and Zn were slightly polluting (Fig 2). In addition, the variation indexes for Mn, As, and Zn were low, indicating that they are evenly distributed in space. These results consistently showed that Mn, As, and Zn levels were relatively unaffected by human factors. Previous studies also showed that the Mn mainly came from the soil parent material [25, 32]. Therefore, factor 3 is considered to be associated with natural sources.

Factor 4 made the highest contribution to Pb at 41.2%. Lead is the main indicator of traffic emissions caused by fuel combustion, and engine and catalyst use [33]. It is estimated that car exhaust emissions account for about two-thirds of global lead emissions [34, 35]. The study area is located near major roads, such as expressways and national highways. Therefore, factor 4 is considered to be a traffic pollution source. In general, the sources of heavy metal pollution in soil are extensive, which means that it is relatively difficult to accurately analyze the sources of heavy metals. This study used the PMF model and correlation analysis to analyze the sources of heavy metals, but did not combine them with the spatial distribution of heavy metal content, which may have led to certain deficiencies.

## 4.2 Main mechanisms associated with the phytoremediation of heavy metal pollution

Phytoremediation uses the aboveground and underground parts of plants to remove and stabilize heavy metals in soil and the main mechanisms are phytoextraction and phytostabilization (Fig 6). Phytoextraction absorbs and transfers heavy metal pollutants to aboveground parts through plant roots and finally removes heavy metals from soil through harvesting and treatment [36–38]. Phytostabilization reduces the mobility and effectiveness of heavy metals by absorbing and accumulating heavy metals in the soil through roots, which improves the stability (harmlessness) of heavy metals in the soil [36–38]. Furthermore, the plant canopy formed after phytoremediation can reduce the near surface wind speed and the diffusion of fine

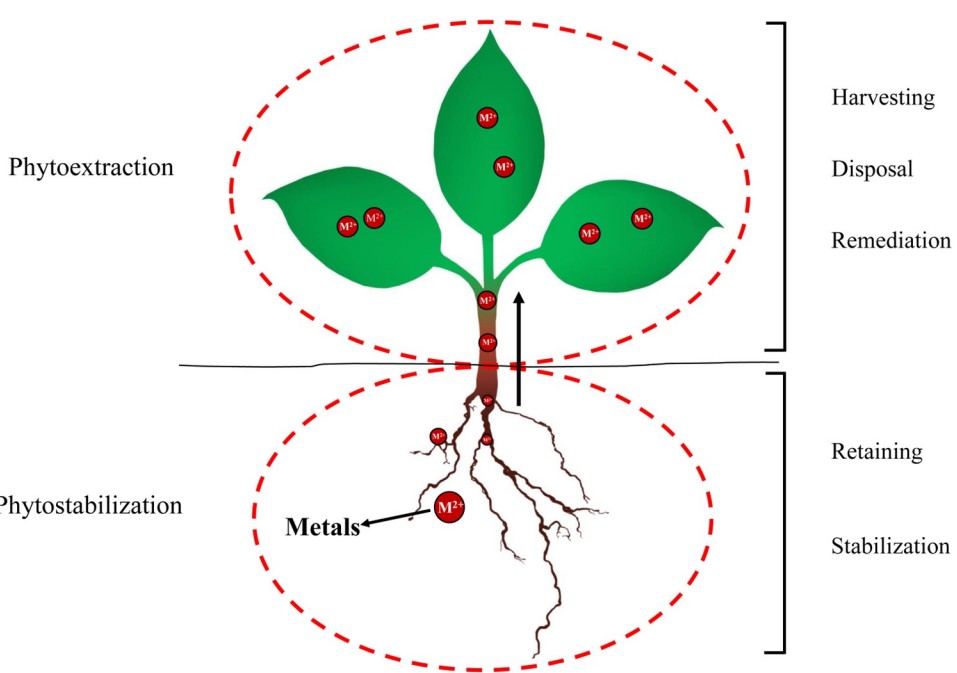

**Fig 6. Phytoextraction and phytostabilization.**

particle pollutants, while the underground root network can prevent rainfall erosion and leaching, and provide a suitable rhizosphere environment for heavy metal precipitation [39, 40].

In general, identifying the natural vegetation growing on contaminated sites and the selection of metal tolerant plants with potential remediation value are some of the most important components of a localized phytoremediation strategy [41]. Mousavi et al. [42] evaluated the remediation potential of plant species growing in a heavy metal polluted saline alkali soil and found that seven plants (*L. arborescens*, *A. santolina*, *P. gnaphalodes*, *Z. eurypteru*, *P. harmala*, *P. olivieri*, and *A. javanica*) have good heavy metal stabilization effects, while *Z. eurypterum* and *A. javanica* also have good heavy metal and relatively high heavy metal extraction abilities, respectively. Liu et al. [43] evaluated the plant stability and phytoextraction potential of native plants naturally growing around tailings ponds in Altay, Xinjiang Province, China. The evaluation results showed that *S. schmidt* was most suitable for stabilizing Cu, *K. caspica* was most suitable for stabilizing Cd, and *P. Aviculare* had a high extraction potential for Cu. Zhao et al. [21] used a fuzzy comprehensive evaluation to evaluate the comprehensive heavy metal extraction potential of woody plants growing on heavy metal contaminated sites and found that *B. papyrifera* could extract many kinds of heavy metals at the same time. In terms of extraction potentials for single heavy metals, this study found that *A. patens* had a strong ability to extract Cr; *P. aviculare* had a strong Zn extraction ability; *S. ruthenica* had a strong Cd extraction ability; *A. breviligulata* had strong Mn, Ni, Cu, As, and Pb extraction abilities. In terms of single heavy metal extraction and stabilization abilities, this study found that *S. ruthenica* had a strong ability to stabilize Cr; *P. harmala* had a strong ability to stabilize Ni, Cu, and Cd; and *A. breviligulata* had a strong ability to stabilize Mn, Zn, As, and Pb. The evaluation results for plant heavy metal comprehensive extraction/stabilization potential based on fuzzy mathematics showed that the plant comprehensive extraction and plant comprehensive stability indexes for *A. breviligulata* were the highest, indicating that *A. breviligulata* had better heavy metal

removal and stabilization effects than the other native plants. Therefore, *A. breviligulata* can be selected as the preferred species for heavy metal pollution remediation in the study area.

## 5. Conclusion

The results showed that there was obvious compound heavy metal pollution in the soil around the copper-nickel tailings reservoir area. Among them, the heavy metal elements Cd, Cu, Ni, and Cr were at the heavy pollution level; Mn and Pb were at the moderate pollution level; and Zn and As were at the slight pollution level. The positive matrix factorization (PMF) model results showed that the Cu and Ni came from industrial pollution sources; Cd and Cr came from atmospheric deposition and agricultural pollution sources; Pb came from traffic pollution sources; and Mn, Zn, and As came from natural sources. This comprehensive evaluation of the remediation potential of several plants showed that the comprehensive enrichment index (*CEI* = 0.81) and comprehensive stability index (*CSI* = 0.83) for *A. breviligulata* were the highest, indicating that it has a strong potential for heavy metal removal and stability in the soil. Therefore, *A. breviligulata* can be used to remediate heavy metal pollution in the soil from the study area. Finally, this study introduces a new evaluation method for phytoremediation potential that may help to find plant species with stronger comprehensive remediation capacities.

## Acknowledgments

The sample analysis and test were conducted in the central laboratory of Xinjiang Institute of Ecology and Geography. We thank Yao Wang and Miao Song for their help in the experiment.

## Author Contributions

**Conceptualization:** Jianfei Shi, Zhengzhong Jin, Zhibin Zhou.

**Data curation:** Jianfei Shi, Wenting Qian.

**Funding acquisition:** Zhengzhong Jin.

**Investigation:** Jianfei Shi, Xin Wang, Xiaoliang Yang.

**Resources:** Wenting Qian, Zhengzhong Jin.

**Writing – original draft:** Jianfei Shi.

**Writing – review & editing:** Jianfei Shi, Zhengzhong Jin, Zhibin Zhou.

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
