## [Decision Letter · Decision Letter 0]

15 Nov 2022

PONE-D-22-29071Heavy metal pollution characteristics of soil around a copper-nickel mine tailings pond in the northwest arid area of China and evaluation of desert phytoremediation potentialPLOS ONE

Dear Dr. Shi,

Thank you for submitting your manuscript to PLOS ONE. After careful consideration, we feel that it has merit but does not fully meet PLOS ONE’s publication criteria as it currently stands. Therefore, we invite you to submit a revised version of the manuscript that addresses the points raised during the review process.

We look forward to receiving your revised manuscript.

Kind regards,

Muhammad Aamer Mehmood, Ph.D.

Academic Editor

PLOS ONE

“This work was funded by the National Key Research and Development Program of China (2018YFC1802903).”

Reviewers' comments:

Reviewer's Responses to Questions

**Comments to the Author**

1. Is the manuscript technically sound, and do the data support the conclusions?

Reviewer #1: Yes

Reviewer #2: Partly

2. Has the statistical analysis been performed appropriately and rigorously? 

Reviewer #1: Yes

Reviewer #2: Yes

3. Have the authors made all data underlying the findings in their manuscript fully available?

Reviewer #1: Yes

Reviewer #2: Yes

4. Is the manuscript presented in an intelligible fashion and written in standard English?

Reviewer #1: Yes

Reviewer #2: No

5. Review Comments to the Author

Reviewer #1: The manuscript entitled “Heavy metal pollution characteristics of soil around a copper-nickel mine tailings pond in the northwest arid area of China and evaluation of desert phytoremediation potential” may be considered for publication after incorporation of suggested changes.

Reviewer #2: The manuscript titled “Heavy metal pollution characteristics of soil around a copper-nickel mine tailings pond in the northwest arid area of China and evaluation of desert phytoremediation potential” is interesting. However, after reading it, I find out that limited information and the following comments may be helpful for the improvement of the manuscript.

1, Abstract: the abstract is too simple and some important information is not fully displayed. Abstract needs to be qualitative as well as quantitative, improve the quality of the abstract with quantitative data.

2, Line 49, G et al., 2015？

3, The introduction is a bit shallow and would benefit of more depth and outlining some hypotheses, the authors need to emphasize the novelty in the introduction part. Otherwise, I believe that this article might be more suitable for local monitoring journals. And the introduction must be focused on the objectives and methods.

4, The quality control parts needs to be more clear: how the standard curves were constructed? what are LoD and LoQ of the methods and how they were determined? Please provide detailed information regarding the accuracy and precision of chemical analysis for every element.

5, National Soil Environment class II standard has been abolished, and you should use a new standard.

6, In the "Results and Discussion" part. The authors lack (or do not show in the manuscript) the deeper knowledge in pollution assessment and source analysis of soil heavy metal. Therefore, I would propose to authors to be better familiar with some recent advances in the field, such as:

An integrated approach to quantifying ecological and human health risks from different sources of soil heavy metals. Science of the Total Environment.

https://doi.org/10.1016/j.scitotenv.2019.134466

An integrated exploration on health risk assessment quantification of potentially hazardous elements in soils from the perspective of sources. Ecotoxicol. Environ. Saf, 208, 15: 111489. https://doi.org/10.1016/j.ecoenv.2020.111489.

Above-mentioned works can give further help about pollution assessment and source analysis of soil heavy metal. It is suggested the authors refer to these studies. Read all of them carefully, please use for improving the manuscript.

7, At the end of the discussion, explore the limitations of the study.

8, Conclusion must be rewritten based on the objectives and the main finding in this research.

9, References: it is highly recommended to use DOI.

10, At some parts of the manuscript, the English language is hard to follow. Please check the English by native speakers.

6. PLOS authors have the option to publish the peer review history of their article (what does this mean?). If published, this will include your full peer review and any attached files.

Reviewer #1: **Yes: **Dr. Sana Ashraf

Reviewer #2: No

---

## [Author Response · Author response to Decision Letter 0]

14 Dec 2022

Dear Editor,

Please find attached the corrected version of our manuscript “Heavy metal pollution characteristics of soil around a copper-nickel mine tailings pond in the northwest arid area of China and evaluation of desert phytoremediation potential”. We appreciate the constructive comments and suggestions from reviewers. These opinions help to improve academic rigor if our article. Based on their suggestion and request, we have made corrected modifications on the revised manuscript. We hope that our work can be improved again. Furthermore, we would like to show the details as follows:

Line number is according to the revised manuscript (clean version)

Response to Reviewer 1 Comments

Point 1：MNV and MTT are not given for all analyzed heavy metals. Please explain.

Response 1: Because some elements have MNV or MTT values that are too large. If displayed in a diagram, it may result in the graph not having aesthetic properties. In order to give the reader an idea of the normal range and toxicity range of each heavy metal element in the plant, I have marked in the analysis. [line 243-259]

Response to Reviewer 2 Comments

Point 1：Abstract: the abstract is too simple and some important information is not fully displayed. Abstract needs to be qualitative as well as quantitative, improve the quality of the abstract with quantitative data.

Response 1: Thanks for your comment, in the new version we have revised the abstract and added some quantitative descriptions to the abstract. [line 23-31]

Point 2: Line 49, G et al., 2015？

Response 2: Thanks for your comment, in the new version we have corrected. [line 45]

Point 3: The introduction is a bit shallow and would benefit of more depth and outlining some hypotheses, the authors need to emphasize the novelty in the introduction part. Otherwise, I believe that this article might be more suitable for local monitoring journals. And the introduction must be focused on the objectives and methods.

Response 3: Thanks for your comment, in the new version we have revised the introduction. We described the objectives of this study and the advantages of evaluation methods. [line 49-73]

Point 4: The quality control parts needs to be more clear: how the standard curves were constructed? what are LoD and LoQ of the methods and how they were determined? Please provide detailed information regarding the accuracy and precision of chemical analysis for every element.

Response 4: The standard curve is prepared by mixing the standard solution and diluting it with 2% nitric acid. The R value of the linear fitting curve of Cu, Ni, Cd, Cr, Zn, Pb and As is greater than or equal to 0.999, and the R value of the linear fitting curve of Mn is greater than or equal to 0.995. LoD and LoQ of the method are calculated as follows:

LoD=(3×sd×A)/B

LoQ=4×LoD

Where sd is the standard deviation of 11 sample blanks, A is the dilution multiple, and B is the weighed volume. Table 1 shows the true and measured values of the plant standard sample GBW 10052a (GSB-30a). Table 2 shows the real and measured values of soil standard sample GBW07426 (GSS-12). The results showed that the measured values of plant samples and soil samples were within the true range of samples, and RSD was less than 5%. These results reflect the precision and accuracy of sample determination to some extent. Thanks for your comment, in the new version we have made it more explicitly stated out. [line 109-112]

Table1 Standard and measured values of GBW 10052a (GSB-30a).

GSB-30a Cr Mn Ni Cu Zn As Cd Pb

Standard value 0.6±0.1 1170±40 4.4±0.3 13.2±0.9 26±3 0.16±0.02 0.2±0.02 1.6±0.2

Measured value1 0.603 1183 4.62 13.7 27.8 0.162 0.215 1.57

Measured value2 0.592 1192 4.62 13.7 27.6 0.157 0.196 1.59

Measured value3 0.613 1194 4.64 14.0 27.7 0.170 0.213 1.60

Measured value4 0.614 1192 4.48 13.6 26.1 0.163 0.218 1.55

Measured value5 0.603 1204 4.49 13.5 26.1 0.159 0.194 1.54

Measured value6 0.598 1195 4.48 13.6 26.5 0.161 0.211 1.58

Measured value7 0.609 1190 4.37 13.1 27.1 0.152 0.198 1.70

Measured value8 0.612 1167 4.30 12.6 26.0 0.165 0.209 1.75

Mean 0.606 1189 4.50 13.4 26.9 0.161 0.207 1.610 

SD 0.008 10 0.12 0.43 0.78 0.005 0.009 0.075 

RSD (%) 1.30 0.91 2.74 3.22 2.89 3.34 4.52 4.65

Table2 Standard and measured values of GBW07426 (GSS-12).

GSS-12 Cr Mn Ni Cu Zn As Cd Pb

Standard value 59±2 774±19 32±1 29±1 78±5 12.2±0.8 0.15±0.02 19±2

Measured value1 57.8 770.3 32.3 30.1 76.7 12.73 0.151 18.6

Measured value2 58.3 765.6 33.0 30.0 77.8 12.87 0.170 18.6

Measured value3 58.6 772.1 32.6 30.1 77.9 12.51 0.160 18.7

Measured value4 59.0 778.4 32.0 29.6 76.9 12.94 0.158 18.6

Measured value5 57.9 764.2 31.4 29.8 73.5 12.91 0.167 18.0

Measured value6 58.2 774.7 31.6 29.5 75.9 12.84 0.166 18.1

Measured value7 58.7 781.5 31.4 28.1 68.3 12.26 0.160 18.6

Measured value8 59.3 776.3 31.5 28.2 68.5 12.32 0.166 18.7

Mean 58.57 772.89 31.99 29.42 74.44 12.67 0.16 18.5 

SD 0.54 6.04 0.59 0.84 3.97 0.27 0.01 0.28 

RSD (%) 0.93 0.78 1.85 2.86 5.33 2.14 3.74 1.51

Point 5: National Soil Environment class II standard has been abolished, and you should use a new standard.

Response 5: Thank you for your comment, in the new version we have changed the reference standard. [line 188]

Point 6: In the "Results and Discussion" part. The authors lack (or do not show in the manuscript) the deeper knowledge in pollution assessment and source analysis of soil heavy metal. 

Response 6: Thank you for your recommended articles, which have a comprehensive analysis of the sources of heavy metals. In the new version, we have conducted a deeper analysis of the sources of heavy metals in the discussion section. [line 308-349]

Point 7: At the end of the discussion, explore the limitations of the study.

Response 7: Thanks for your comment, in the new version we discussed the shortcomings of this study. [line 351-353]

Point 8: Conclusion must be rewritten based on the objectives and the main finding in this research.

Response 8: Thanks for your comment, in the new version we rewrote our conclusions based on the objectives and the main finding in this research. [line 394-406]

Point 9: References: it is highly recommended to use DOI.

Response 9: Thanks for your comment, in the new version we added the DOI number of references.

Point 10: At some parts of the manuscript, the English language is hard to follow. Please check the English by native speakers.

Response 10: Thanks for your comment. The submission has been checked by the editor whose native language is English.

---

## [Decision Letter · Decision Letter 1]

2 Jan 2023

Evaluation of soil heavy metals pollution and the phytoremediation potential of copper-nickel mine tailings ponds

PONE-D-22-29071R1

Dear Dr. Jianfei Shi

We’re pleased to inform you that your manuscript has been judged scientifically suitable for publication and will be formally accepted for publication once it meets all outstanding technical requirements.

Kind regards,

Muhammad Aamer Mehmood, Ph.D.

Academic Editor

PLOS ONE

Reviewer's Responses to Questions

**Comments to the Author**

1. If the authors have adequately addressed your comments raised in a previous round of review and you feel that this manuscript is now acceptable for publication, you may indicate that here to bypass the “Comments to the Author” section, enter your conflict of interest statement in the “Confidential to Editor” section, and submit your "Accept" recommendation.

Reviewer #1: All comments have been addressed

Reviewer #2: All comments have been addressed

2. Is the manuscript technically sound, and do the data support the conclusions?

Reviewer #1: Yes

Reviewer #2: Yes

3. Has the statistical analysis been performed appropriately and rigorously? 

Reviewer #1: Yes

Reviewer #2: Yes

4. Have the authors made all data underlying the findings in their manuscript fully available?

Reviewer #1: Yes

Reviewer #2: Yes

5. Is the manuscript presented in an intelligible fashion and written in standard English?

Reviewer #1: Yes

Reviewer #2: Yes

6. Review Comments to the Author

Reviewer #1: The manuscript entitled “Evaluation of soil heavy metals pollution and the phytoremediation potential of coppernickel mine tailings ponds” is of good quality and written in a good way. The manuscript has been revised carefully by the authors. The authors have incorporated all the suggested corrections in the manuscript. The current revised manuscript may be considered for publication.

Reviewer #2: The manuscript has been properly revised and improved, and I recommend the publication of it as it is.

7. PLOS authors have the option to publish the peer review history of their article (what does this mean?). If published, this will include your full peer review and any attached files.

Reviewer #1: No

Reviewer #2: No

---

## [Editor Report · Acceptance letter]

11 Jan 2023

PONE-D-22-29071R1 

Evaluation of soil heavy metals pollution and the phytoremediation potential of copper-nickel mine tailings ponds 

Dear Dr. Shi:

I'm pleased to inform you that your manuscript has been deemed suitable for publication in PLOS ONE. Congratulations! Your manuscript is now with our production department. 

Kind regards, 

on behalf of

Dr. Muhammad Aamer Mehmood 

Academic Editor

PLOS ONE